# Acceptability of self-sampling for high-risk HPV DNA testing for primary cervical cancer screening among women in Thyolo, Malawi: A qualitative study

Hussein H. Twabi[1,2,3]◉*, Wakumanya Sibande[1]◉, Owen Mhango[1], Takondwa Charles Msosa[1,4,5], Chikumbutso Chipandwe[1], Madalo Mukoka[1,6], Moses Kumwenda[1,3], Vanitha Sivalingam[7], Dennis Solomon[8], Chisomo Msefula[1], David Lissauer[2,3], Marriott Nliwasa[1]‡, Maria Lisa Odland[2,3,9,10]‡

1 Kamuzu University of Health Sciences, Blantyre, Malawi, 2 Institute of Life Course and Medical Sciences, University of Liverpool, Liverpool, United Kingdom, 3 Malawi-Liverpool-Wellcome Trust Programme, Blantyre, Malawi, 4 Amsterdam UMC, location University of Amsterdam, Department of Global Health, Amsterdam, The Netherlands, 5 Amsterdam Institute for Global Health and Development, Amsterdam, The Netherlands, 6 Department of Infectious Disease Epidemiology and International Health, London School of Hygiene and Tropical Medicine, London, United Kingdom, 7 Department of Obstetrics and Gynaecology, Liverpool Women's Hospital NHS Foundation Trust, Liverpool, United Kingdom, 8 Thyolo District Hospital, Ministry of Health, Thyolo, Malawi, 9 Department of Public Health and Nursing, Norwegian University of Science and Technology, Trondheim, Norway, 10 Department of Obstetrics and Gynaecology, St. Olavs University Hospital, Trondheim, Norway

◉ HHT and WS should be considered joint first authors
‡ MLO and MN should be considered joint last authors
* husseintwabi@hotmail.com

## Abstract

Despite the roll-out of cervical cancer screening within routine health facilities, uptake of cervical cancer screening in Malawi remains low due to factors such as privacy concerns, stigma, and distance to health facility. Self-sampling for human papillomavirus (HPV) testing offers a viable alternative to provider-delivered sampling, resolving issues with accessibility and privacy related barriers. However, there is limited understanding of the acceptability of self-sampling among women in low-income settings. We aimed to assess women's perceptions, experiences, and acceptability of self-sampling for HPV testing for primary cervical cancer screening. We conducted a qualitative phenomenological study among ten purposively sampled women screening for cervical cancer at a rural hospital in Malawi. Data collection instruments and the thematic analytical approach were guided by the Theoretical Framework of Acceptability. Key constructs that were central to this analysis included affective attitude, burden, self-efficacy, intervention coherence, and perceived effectiveness. Self-sampling emerged as a culturally and socially acceptable diagnostic practice by the women, mediated by emic values of privacy, bodily autonomy, and convenience. Women valued the ability to autonomously collect samples, which allowed them to circumvent discomfort and perceived invasiveness linked to speculum use, especially by male clinicians. Sociocultural barriers

**Data availability statement:** Data are uploaded to Zenodo repository, DOI: https://doi.org/10.5281/zenodo.15876856.

**Funding:** The conduct of this study was made possible by the PhD research funds for the corresponding author, HHT, who is supported by a Helse Nord RHF grant to the Kamuzu University of Health Sciences. DL is supported by a National Institute for Health and Care Research (NIHR) Global Health Professorship (NIHR300808), using UK aid from the UK Government to support global health research. The funders had no role in study design, data collection, data analysis, data interpretation, writing of the report, or the decision to submit this paper for publication.

**Competing interests:** The authors have declared that no competing interests exist.

such as limited literacy levels, poor access to transport, and a lack of spousal approval restricted broader acceptability. Facilitators of acceptability included comprehension of the intervention, confidence in the self-sampling process, and the potential of the intervention to increase screening uptake, reduce cervical cancer screening-related stigma and reduce healthcare worker burden. Self-sampling for cervical cancer screening is an acceptable and promising alternative for improving the uptake of screening among women in Malawi and similar low-income countries. Scale-up of this approach will require addressing socio-cultural barriers through optimising instructional materials, engaging male partners, and leveraging community health workers for scaled community implementation via task-sharing.

## Introduction

Cervical cancer ranks as the fourth most prominent cause of female cancer cases worldwide, contributing significantly to cancer-related fatalities in women.[1] It is particularly devastating in low- and middle-income countries (LMICs), with sub-Saharan Africa bearing the highest burden.[1] In sub-Saharan Africa in 2020, there were approximately 117,944incident cases of cervical cancer, along with 76,140associated deaths, which translates to standardised rates of 33.4 new cases and 22.6 deaths per 100,000 individuals, respectively.[1]

Malawi stands out as having the world's highest recorded rates of cervical cancer incidence and mortality. With an age-standardised incidence rate of 67.9 per 100,000 and an age-standardised mortality rate of 51.5 per 100,000, the country's burden of disease is double the sub-Saharan African estimates.[2] Additionally, there is a rising trend in the number of new cervical cancer cases in the country.[3] The burden of cervical cancer in Malawi is further aggravated by its association with HIV, as the disease is strongly linked to the prevalence of HIV regardless of antiretroviral therapy.[4–6]

Although cervical cancer is preventable and treatable, actual screening and management efforts in Malawi fall short. In Malawi, the cervical cancer screening program is based on the screen-and-treat approach, which involves visual inspection with acetic acid (VIA) and cryotherapy or thermocoagulation.[7] Despite VIA's cost-effectiveness and immediate lesion management advantages, it suffers from variability in interpretation and reduced sensitivity in older individuals, necessitating quality control measures.[8–10] Notwithstanding national VIA rollouts, coverage remains below targets, compounded by high loss-to-follow-up rates.[7,11]

Human papillomavirus (HPV) deoxyribonucleic acid (DNA) testing is more effective in screening for cervical cancer than conventional and liquid-based cytology.[12] HPV DNA testing was considered to be not feasible in low-resource settings due to requirements of expensive laboratory infrastructure and the long processing time.[12] Emerging lower cost test for HPV DNA are making it possible to implement HPV testing in low-resource settings, and this may be the best strategy for cervical cancer screening in such settings.[12]

There have been recent efforts to establish novel screening modalities that use HPV DNA testing instead of VIA for screening of cervical cancer. In Malawi, implementing partners such as Elizabeth Glaser Paediatric AIDS Foundation (EGPAF) have begun the rollout of routine testing for HPV DNA using Cepheid Gene Xpert, leveraging the pre-existing Cepheid Gene Xpert resources put in place in response to the TB epidemic in the country. However, there have been challenges in consistent supply of cartridges for HPV DNA testing, coupled with the competing need for the use of the existing Cepheid Gene Xpert machines. Consequently, most patients are deprived from having access to HPV DNA testing, and those who do, experience long turnaround times for Xpert HPV results, essentially negating the efforts of the screen-and-treat strategy.

The challenges in implementing HPV DNA testing in Malawi underscore the urgent need for alternative screening strategies that address existing barriers. Self-sampling for HPV testing offers a practical and scalable solution to mitigate bottlenecks associated with limited Cepheid Gene Xpert resources.[13] By enabling women to independently collect samples, self-sampling dramatically reduces dependence on clinical infrastructure and alleviates the burden on overextended diagnostic systems. Additionally, self-sampling facilitates community-based or home-based sample collection, which can improve access for women in remote or underserved areas.[13]

Besides screening modalities, various other barriers to accessing care exist in low-resource settings that result in poor uptake of cervical cancer screening services. These include perceptions (confusions about why the need for screening when asymptomatic), difficulty in navigating health care facilities, lack of information regarding direction of where and when to obtain service, distance to the health facilities, as well as stigma and suspicions about the HIV status of women accessing cervical cancer screening, as most cervical cancer screening services are delivered by HIV care and treatment units in Malawi.[14] Self-sampling has the potential to address these barriers by providing a private, convenient, and less invasive alternative, particularly for those in remote areas or with limited access to healthcare facilities. However, its success depends on the willingness and comfort of women to adopt this approach. We thus conducted this study to explore the acceptability of women to undergo screening for cervical cancer using self-sampling as opposed to clinician-sampling for near point-of-care (POC) high-risk HPV DNA testing.

## Methods

### Ethical statement

This study was approved by the College of Medicine Research and Ethics Committee (COMREC) (approval number: P.05/24–0759). The District Health Research Committee provided permission to conduct the study at Thyolo District Hospital. All participants provided written informed consent to participate in the study. In accordance with COMREC regulations, all participants received $10 as compensation for their time given to a broader feasibility study as well as this qualitative component.

### Study design and setting

This study was conducted as part of a broader study investigating the feasibility of a novel HPV self-sampling kit coupled with a POC rapid testing kit for primary HPV DNA testing for cervical cancer screening at Thyolo District Hospital (TDH), a secondary level healthcare facility in Thyolo District in south-western Malawi. We employed a phenomenological design to understand the acceptability and willingness of women to participate in cervical cancer screening using self-sampling as opposed to clinician-sampling for POC HR-HPV DNA testing. The phenomenological design allowed us to deeply explore the lived experiences of women accessing cervical cancer screening concerning the use of the intervention and to define the implementation context of the intervention.[15]

### Description of the intervention

The intervention, the Delphi Vaginal self-sampler, [16] is a user-friendly tool designed for women to self-collect cervical samples for HPV testing, aiding in cervical cancer screening. It features a shaft for insertion, a pre-measured sterile buffer

solution for lavage, and an integrated collection container that securely stores the sample. The device is easy to use, requiring insertion, flushing with lavage fluid, and withdrawal to collect exfoliated cervical cells. A leak-proof vial ensures sample integrity during transport to a laboratory. The sample collected can be stored at ambient room temperature for up to seven days without losing its integrity. This non-invasive and privacy-respecting tool empowers women to participate in screening programs, particularly in resource-limited or high-stigma environments, enhancing accessibility and early detection efforts.

## The theoretical framework of acceptability

Acceptability is a complex concept that measures how appropriate a healthcare intervention is viewed by both providers and recipients. It depends on the anticipation and the cognitive and affective reactions that the intervention evokes.[17] By reflecting on the actual and perceived experiences of providers and recipients, the Theoretical Framework of Acceptability (TFA) facilitates the evaluation of intervention acceptability.[17] As a result, the TFA encompasses more than just the elements of the intervention. It also considers organisational, personal, and wider contextual elements that could affect how the intervention is delivered and whether it is acceptable.[17,18] The TFA consists of seven constructs (shown in Table 1), that assess acceptability before, during, and after the intervention has been delivered.[17] This framework has been widely used in assessing the intervention recipient's acceptability of several health interventions including preventive, [18] curative, [19] and disease management in primary care.[20] We used the TFA in this study to help us identify key factors that could influence women's acceptability of self-sampling for cervical cancer screening (Table 1).

## Research team

The primary research team consisted of two investigators, HHT (MBBS, MSc) and OM (BSoc), both trained in qualitative research methods and fluent in Chichewa (the local language) and English. HHT is a male physician epidemiologist, while OM is a male specialist in social and behavioural research in health. Collectively, the team of investigators had more than five years of experience in conducting qualitative research. No prior relationships existed between the researchers and participants before the study. To enhance validity, trustworthiness and ethical rigor of the study, all the researchers engaged in reflexivity throughout data collection and analysis by maintaining reflective journals.

## Study population and recruitment

This study purposively recruited 10 women from a larger feasibility study that was evaluating the use of a novel HPV self-sampling kit coupled with a POC rapid testing kit for HPV DNA testing. Eligible participants were women residing in the catchment area of Thyolo District Hospital who had previously undergone cervical cancer screening using clinician-collected samples tested with Xpert HPV. As part of the feasibility study, each participant was provided with a self-sampling kit, received instruction on how to collect a cervicovaginal sample, and was directed to a private space to

**Table 1. Constructs and descriptions of the theoretical framework of acceptability.**

| Construct | Description |
| --- | --- |
| Affective attitude | How an individual feels about the intervention |
| Burden | The perceived amount of effort that is required to participate in the intervention |
| Ethicality | The extent to which the intervention has a good fit with an individual's value system |
| Intervention Coherence | The extent to which the participant understands the intervention and how it works |
| Opportunity Costs | The extent to which benefits, profits, or values must be given up to engage in the intervention |
| Perceived Effectiveness | The extent to which the intervention is perceived as likely to achieve its purpose |
| Self-efficacy | The participant's confidence that they can perform the behaviour(s) required to participate in the intervention |

This description has been adopted from Sekhon, Cartwright & Francis (2017) [18]

perform the collection. The self-collected samples were then submitted to the research team for later batch processing with the POC rapid test. The purposive approach was preferred to ensure heterogeneity in terms of age, marital status and HIV status. Saturation was considered to be achieved by the 10th interview, as no new themes were emerging and there was noticeable repetition of ideas and perspectives across participants. This was verified through daily debriefings and iterative analysis conducted during data collection. Women aged 18 and above who attended the clinic were eligible to participate in the study. Despite the Malawi National cervical cancer screening guidelines recommending the age at initial screening to be 25 years, [21] younger women still seek the screening services and are not denied the service by the facility. No person who was approached by the study team refused to participate in the interviews. The decision to use only in-depth interviews (IDIs) in this qualitative study was guided by the need to explore individual experiences, perceptions, and decision-making processes in a private and detailed manner.

Recruitment was facilitated by a research assistant (CC) at the cervical cancer screening unit at Thyolo District Hospital, who was already known to the participants. The research assistant introduced the broader feasibility study to the participants during routine health talk sessions and those that were willing to participate in the study were invited to a private area for the informed consent process.

## Data collection

The data for this study was collected between 10 October and 16 November 2024. The interviews were conducted face-to-face in Chichewa by OM, a graduate and experienced qualitative research who received a protocol training specific to the current study. The interviewer used a pre-tested interview guide with open-ended questions and probes aiming to explore key issues: Experiences and Perceptions of Self-sampling for Cervical Cancer; Factors Enhancing User Confidence in Self-sampling Self-efficacy; and Enabling Factors for Self-Sampling Adoption Perceived Effectiveness.

IDIs lasted an average of one hour and were recorded using digital audio recorders and field notes containing summaries for each interview. Daily briefing meetings were held with HHT to discuss emerging concepts or themes and reflect on the influence of the researchers during the data collection process. This reflexive and iterative process helped to improve the data collection process and enrich the collected data during fieldwork. Observable through these briefing meetings, there was evidence of data saturation as recurring responses and consistent perspectives were apparent across participants. Transcripts were not returned to study participants for comments or clarifications, and no repeat interviews were conducted.

## Data analysis

Once all interviews had been completed, audio files were transcribed verbatim and directly translated into English by OM. Each transcription had a unique identification code that was assigned to the participant during the interview. Data were coded using inductive coding and organised using Nvivo. OM familiarised himself with the data by reading and re-reading before conducting initial coding to identify preliminary codes, which were subsequently reviewed and discussed with HHT to refine the coding framework. To ensure intercoder reliability, WS, an independent coder, coded the transcripts to collaborate with the identified codes. A coding tree was developed iteratively during analysis, beginning with inductive open codes which were then grouped into categories and mapped onto five constructs of the TFA: affective attitude, burden, self-efficacy, intervention coherence, and perceived effectiveness. Following this, WS identified patterns across the coded data and developed initial themes. These themes were then iteratively reviewed, split into sub-themes, regrouped, and merged where necessary to generate broader themes that encapsulated the key findings of the study. The final broader themes were then mapped onto the TFA [18]. The TFA consists of seven constructs: affective attitude, burden, ethicality, intervention coherence, opportunity costs, perceived effectiveness, and self-efficacy. Five constructs (affective attitude, burden, intervention coherence, perceived effectiveness, and self-efficacy) meaningfully emerged from our inductive thematic analysis. This final mapping facilitated a structured analysis and interpretation of the data within a theoretical framework.

We used Thematic Analysis (TA) to analyse our data. TA was chosen due to its inherent flexibility, accommodating multiple approaches to interpret meaning. We used TA to map out patterns and relationships across our data that focused on the perceptions and experiences of our participants.[22] We used relevant quotes selected from the data to illustrate the identified key themes. The Consolidated Criteria for Reporting Qualitative Research (COREQ) checklist was followed while reporting our findings.

## Findings

The study included ten female participants recruited at Thyolo District Hospital who received cervical cancer screening using Xpert HPV, ranging in age from 19 to 47 years, with most women being between 31 and 39 years old (n = 7). The majority had attained primary school education (n = 7), with all participants being able to read a newspaper or letter. The majority of participants (n = 6) self-reported as living with HIV. Characteristics of participants are detailed in Table 2 below.

Our findings identified three major themes which mapped to five TFA [17] constructs: Perceptions and experiences with self-sampling for cervical cancer screening (Affective attitude and Burden), Factors enhancing user confidence in self-sampling for cervical cancer screening (Self-efficacy and Intervention coherence) and Enabling factors for self-sampling adoption (Perceived Effectiveness). Fig 1 summarises the responses of participants mapped to the TFA constructs that emerged.

Each construct is linked to specific facilitators or barriers emerging from participant narratives. The arrows indicate potential directional influences among constructs, illustrating how understanding the intervention (intervention coherence) can enhance confidence (self-efficacy), while burden and perceived effectiveness can impact overall acceptability both positively and negatively. The model highlights the complex interplay of emotional, cognitive, and contextual factors shaping women's acceptance of self-sampling in low-resource settings.

### Experiences and perceptions of self-sampling for cervical cancer

**Affective attitude.** This construct describes how individuals feel about a particular intervention [17]. Overall, women's perceptions and experiences of self-sampling for cervical cancer screening were generally favourable, indicating that the intervention was highly acceptable. Privacy concerns were one of the main reasons why self-sampling was preferred over provider-delivered sampling. A majority of women narrated that because provider-delivered sampling compelled them to expose their bodies, it violated their privacy, especially when male clinicians carried it out:

*"I would say that self-sampling for cervical cancer offers privacy to women because with self-sampling you can self-sample without the presence of anyone, a thing that can make one get rid of the shyness that comes when a male clinician is collecting the sample."* HPVF-004

**Table 2. Characteristics of the participants of the IDIs.**

| Participant number | Age | Highest education level | Can read a newspaper or letter | Employment status | Self-reported health status | Marital status | Self-reported HIV status |
|---|---|---|---|---|---|---|---|
| HPVF-002 | 30 | Never been to school | No | Unemployed | Good | Married | Positive |
| HPVF-004 | 27 | Primary school | Yes | Piece worker | Very good | Married | Positive |
| HPVF-006 | 33 | Primary school | Yes | Self-employed | Good | Married | Negative |
| HPVF-008 | 41 | Secondary school | Yes | Self-employed | Good | Married | Positive |
| HPVF-010 | 19 | Primary school | Yes | Piece worker | Very good | Married | Negative |
| HPVF-013 | 47 | Primary school | Yes | Self-employed | Very good | Widowed | Positive |
| HPVF-027 | 35 | Primary school | Yes | Self-employed | Poor | Married | Positive |
| HPVF-033 | 39 | Secondary school | Yes | Self-employed | Good | Married | Positive |
| HPVF-041 | 34 | Primary school | Yes | Piece worker | Poor | Polygamous marriage | Negative |
| HPVF-042 | 31 | Primary school | Yes | Unemployed | Fair | Polygamous marriage | Positive |

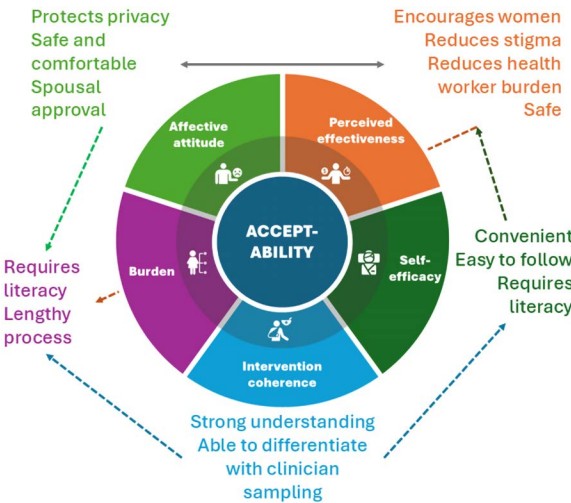

**Fig 1. Conceptual framework illustrating constructs from the Theoretical Framework of Acceptability (TFA) [17] that influence the acceptability of self-sampling for cervical cancer screening.**

Women's positive experiences with self-sampling, especially its capacity to protect their privacy, have the potential to significantly increase women's involvement and uptake of cervical cancer screening. One participant brought this to light, saying:

*"Like I said, this method is simple and easy. Women who are discouraged from coming to the hospital because of shyness would be able to use this method because they will be able to self-sample privately." HPVF-008*

Safety and comfort that comes with self-sampling kits were also recognised and appreciated by women in this study, particularly among a majority of women who experienced pain caused by the speculum during provider-delivered sample collection:

*"Unlike the one that is done at the hospital by the clinicians, self-sampling for cervical cancer is not painful like provider-delivered sampling which involves the insertion of a speculum into the vagina which is very painful as the speculum is made up of the metal." HPVF-006*

Another barrier to cervical cancer screening that self-sampling may overcome was reported to be a lack of support and approval from the spouse. Many participants reported that men often discourage their women from engaging in health-seeking activities, such as permitting and encouraging them to get screened. Spousal support can take different forms, such as paying for transport or accompanying their wives to the hospital. Additionally, some husbands were reported to be reluctant to let their wives access screening services due to feelings of jealousy, as they were uncomfortable with their wives being examined by male clinicians.

*"… some men discourage their wives to go and test for cervical cancer at the hospital for fear that their wives will be tested by a male clinician." HPVF-004*

## Experiences and perceptions of self-sampling for cervical cancer

**Burden of self-sampling for cervical cancer.** The perceived amount of work necessary to take part in an intervention is referred to as burden.[17] Given the low literacy levels among most women in rural areas, participants expressed concerns that most women might struggle to follow sample collection procedures correctly:

*"I would say that some people who are illiterate cannot take the sample correctly and some of the people from the village cannot correctly interpret their test results." HPVF-004*

Beyond literacy challenges, women who participated in the intervention emphasised the time burden associated with self-sampling for cervical cancer screening. This included the time spent on sample collection, traveling to the hospital to submit the sample, and waiting for the sample to be processed:

*"Aaaah I would say that the thing that I didn't like about self-sampling for cervical cancer is that they told us that after self-sampling we should take the sample to the hospital for testing, so I feel like this is a long process. It is different from the provider-delivered testing, and one gets results right there." HPVF-006*

Another participant described the lengthy process associated with self-sampling for cervical cancer screening:

*"Yes, it will take a lot of time for one to self-sample at home let's say then go to the hospital with the sample so that it should be tested by the clinician. It is very slow as compared to provider-delivered sampling where the test is done immediately." HPVF-004*

Recognising the lengthy process associated with self-sampling, some women suggested integrating self-sampling with self-screening and result interpretation to improve convenience and accessibility:

*"I would love it if women would be given the testing device so that they should also self-test for cervical cancer after self-sampling. After self-sampling, one should be able to test and interpret her results." HPVF-006*

## Factors enhancing user confidence in self-sampling

**Self-efficacy.** Self-efficacy refers to participants' confidence in their ability to perform the activities and behaviours necessary for engaging in the intervention [17]. The majority of the participants demonstrated a relatively high level of confidence in the self-sampling process. This was evident in their comprehension of the procedure, and willingness to recommend the intervention to others. One woman narrated her understanding of the procedure and its convenience:

*"With self-sampling, we can collect the sample while at home and bring it to the hospital after getting done with household chores rather than coming to the hospital very early to get screened." HPVF-013*

Participants' preference between provider-sampling and self-sampling also demonstrated their confidence. When asked about their preference, one participant said this:

*"I will choose self-sampling because the method is very easy. It is as easy as inserting the penis into the vagina and I don't think there is any problem that women can experience with self-sampling for cervical cancer." HPVF-002*

Participants in this study exhibited high confidence in the intervention and its procedures, with many expressing their willingness to encourage other women to participate:

> *"I will encourage other women to come and test for cervical cancer using the self-sampling method and with this method, women can come and get this tool and self-sample for cervical cancer at their respective homes and bring the sample here at the hospital so that the doctors can conduct the test. I would recommend it to someone because it is easy and simple." HPVF-042*

However, although women demonstrated confidence in the self-sampling procedure, some participants expressed concerns that could negatively impact users' confidence. These concerns include low literacy levels, the time burden, and a lack of spousal approval and support:

> *"Aah, I would say that there would not be a lot of problems maybe it could be that some people would not be able to self-sample for cervical cancer screening if they do not understand the instructions properly because when one does not understand the instructions clearly and conduct the process in a wrong way, she cannot trust her test results and is going to think that the kit has problems. But it would only require that one should have a thorough understanding of the device before using it." HPVF-008*

### Intervention coherence

This construct encompasses the participants' understanding of the intervention, including its description and how it works [17]. Participants expressed a relatively strong awareness of self-sampling for cervical cancer screening as they were able to provide a standard description of the intervention like this participant:

> *"Self-sampling is when you can comfortably conduct the sampling process yourself without anyone seeing your private parts. With this method, women can come and get this tool and self-sample for cervical cancer at their respective homes and bring the sample here at the hospital so that the doctors can conduct the test. I would recommend it to someone because it is easy and simple." HPVF-002*

Participants were also able to differentiate between routine sampling and self-sampling procedures:

> *"Mmm like I said earlier the self-sampling tool is made up of a plastic kit, and the other thing is that I would be able to conduct the process on my own, meaning that I would be comfortable doing that as compared to the provider delivered sampling where a metal-like swab is inserted into the vagina." HPVF-006*

### Enabling factors for self-sampling adoption

**Perceived effectiveness.** Perceived effectiveness is the degree to which an intervention is thought to accomplish its intended purpose. [17] Individuals' opinions about how beneficial the intervention is can influence their willingness to adopt and adhere to it, shaping overall engagement and outcomes. The majority of participants perceived that the intervention may increase women's cervical cancer screening uptake. One of the participants highlighted the potential self-sampling has to encourage women who are typically shy or apprehensive about clinician-administered screening:

> *"I liked this new self-sampling method more than clinician sampling because even those women who are afraid to be screened by clinicians can accept the self-sampling method." HPVF-013*

Furthermore, participants from this study perceived self-sampling as a strategy to reduce healthcare worker workload by minimizing the need for provider-delivered screening:

> *"It can reduce the workload of the doctors. This is because the doctor can only conduct the test when you self-sample and they will not spend a lot of time like the one that they use here at the hospital." HPVF-008*

An interesting perceived benefit of the intervention is its potential to reduce cervical cancer-related stigma within communities. Participants noted that women seeking cervical cancer services often face stigma, as the screening is commonly associated with being HIV-positive:

> *"As I said earlier if self-sampling for cervical cancer can be adopted, most women will prefer it because of the challenges that women experience when they go to test for cervical cancer at the hospital because most people think that women who come to seek cervical cancer services here at the hospital are infected with HIV. So self-sampling sampling for cervical cancer can help to reduce the stigma among women who seek cervical cancer services." HPVF-002*

Self-sampling is also perceived to be safer compared to provider-administered sampling as the procedure is not painful:

> *"I would say that self-sampling for cervical cancer is very safe. It is different from provider-delivered sampling since it is not painful, and it can encourage women to test for cervical cancer." HPVF-005*

## Discussion

To our knowledge, this is the first qualitative study exploring the acceptability of women to conduct self-sampling for high-risk HPV DNA testing for primary cervical cancer screening in low-income settings. The key findings of this study indicate that self-sampling for high-risk HPV DNA testing for primary cervical cancer screening is widely accepted and has the potential to increase cervical cancer screening uptake and reduce cervical cancer-related complications. In our study, we found that women particularly appreciated the level of privacy, comfort, and convenience of self-sampling compared to a provider-sampling. We established that self-sampling provides women the liberty to collect the sample at a place and time of their convenience, addressing concerns related to discomfort with healthcare providers – especially male clinicians – and pain associated with speculum use. Self-sampling is also perceived as a solution to healthcare accessibility barriers, such as transportation challenges. The study also unearthed several challenges, that if addressed, could facilitate the seamless adoption of self-sampling for cervical cancer screening. These challenges include low literacy levels, lack of spousal approval, and prolonged waiting times for sample submission at the health facility. Furthermore, participants' confidence in utilizing the intervention and their perception of its effectiveness were identified as significant factors influencing both user confidence and the adoption of self-sampling for cervical cancer screening.

Women expressing their preference for self-sampling over provider-delivered sampling due to privacy concerns suggest that perceived invasiveness and discomfort associated with clinician-administered screening contribute to lower uptake of cervical cancer services. This finding is consistent with research in both high- and low-income settings, which found high overall acceptability of self-sampling for cervical cancer and improved outcomes in cervical cancer services. [13,23–30] A quantitative study conducted in Guatemala concluded that self-sampling was highly accepted, with 81% of women considering the method comfortable.[26] Similarly, studies in Canada and Cambodia also reported high acceptability, citing convenience, comfort, privacy, and ease of use as key contributing factors.[25,31] This finding underscores the need for the full-scale adoption of self-sampling for cervical cancer screening, which has been previously suggested in a study exploring women's experiences with a community-based screen-and-treat cervical cancer prevention intervention in Malawi.[32]

Despite the positive perception of self-sampling, challenges exist, including low literacy levels and time constraints. These barriers have significant implications for the wider implementation of the intervention, and are consistent with other studies that found comparable challenges in implementing self-sampling for cervical cancer screening.[33,34] This contrasts the findings of Sormani et al. who reported that the preference for clinician sampling was strongly associated with attending secondary education or higher.[35] Despite this, the same study still reported that the main reported reason for women preferring clinician-sampling was a lack of "self-expertise", [35] which is often discussed in terms of self-efficacy, and has been shown to be associated with education levels.[36–40] Low literacy levels could result in incorrect use of self-sampling kits and erode confidence in the results. The challenge of low literacy levels underscores the need for cognitive studies of Instructions for Use (IFU) to make sure that they are tailored to the cognitive abilities of users. A similar technique has been effective in refining IFUs for HIV self-testing kits in Malawi, Zambia, and Zimbabwe [41].

Self-sampling for cervical cancer screening has the potential to significantly increase the number of women screened by decentralizing the sample collection process and removing the need for a clinician to be present. By allowing women to collect samples themselves – either at home or in community settings – self-sampling reduces reliance on limited clinical infrastructure, frees up healthcare workers' time, and enables task-sharing with community health workers. This shift not only expands coverage but also alleviates the burden on already overstretched health systems, particularly in low-resource settings. Additionally, by reducing clinic congestion and enabling more efficient use of diagnostic platforms, self-sampling contributes to a more streamlined and scalable screening program. However, women recognised barriers like the time needed to gather and carry samples to the hospital, and waiting for the sample to be collected, which could discourage participation, even if they valued the privacy and comfort offered by the self-sampling approach. To address this, the approach can be scaled with complementary investments in sample transport – such as engaging with riders for health (R4H), [42] decentralised sample collection hubs – as is the case with sputum collection points being used in the TB programme and the innovative use of the community women groups such as Village Banking groups, and digital platforms for results dissemination. Additionally, institution of task-sharing of sample collection and results compilation with community-based healthcare workers such as Health Surveillance Assistants (HSAs) may promote the uptake of self-sampling. Task-sharing has been proven to be an effective strategy in addressing screening, diagnosis, and treatment gaps in communicable and non-communicable infections at the community level.[43] Furthermore, it is equally important to establish clear referral pathways to ensure that women with abnormal results are promptly linked to confirmatory testing and treatment, possibly through similar integration with existing community health structures and follow-up support by HSAs.

Another significant barrier to cervical cancer screening was found to be spousal disapproval, as some males are reported to be hesitant to support their spouses or have jealous feelings over their spouses being examined by male clinicians. This finding is consistent with numerous studies highlighting the influence of male partners on women's healthcare decisions, particularly in culturally conservative settings, where male partners often hold significant decision-making power regarding their spouses' healthcare, and their negative attitudes can lead to a direct refusal of screening.[44–47] The fear of social stigma and potential marital discord further compounds these issues, making women reluctant to pursue screening against their partners' wishes. Interventions aimed at increasing cervical cancer screening uptake must consider the crucial role of male partners and engage them in educational programs to address misconceptions and foster supportive attitudes.

Women in this study demonstrated a clear understanding and awareness of self-sampling evidenced in their ability to describe key characteristics and distinguish it from provider-delivered sampling. Their confidence in conducting the procedure was further reflected in their willingness to self-collect samples for cervical cancer screening. This suggests that self-sampling for cervical cancer screening is widely acceptable among women. This is consistent with findings from other studies on self-sampling for cervical cancer screening, which found that preference and uptake were positively influenced by strong knowledge and awareness of self-sampling and high self-confidence in sample collection.[23–25,34]

Conversely, a lack of confidence in sample collection was found to be a barrier to self-sampling for HPV screening, [48] underscoring the necessity of customised interventions to overcome possible hesitancy and boost user confidence.

Lastly, we explored the perceived effectiveness of self-sampling for cervical cancer screening as a key determinant of its acceptability among women in Malawi. Understanding how women view the intervention's capacity to fulfil its intended goal is crucial for informing strategies to improve uptake and integration into routine cervical cancer screening programs. Our findings on perceived effectiveness are consistent with other studies from China, [49,50] Ethiopia, [33] and Kenya, [51] which have also demonstrated the effectiveness of self-sampling for cervical cancer screening. Self-sampling is a promising strategy to increase the adoption of cervical cancer screening, especially among women who are reluctant to seek clinical care because of stigma, privacy concerns, or fear of discomfort. Additionally, self-sampling is also considered as a potential means to lessen healthcare worker workload. To maximise this benefit, task-sharing approaches could be investigated for sample collection, results compilation, and follow-ups for individuals who test positive for high-risk HPV. This model has shown promise in high-income countries like Canada, where community champions encourage self-sampling efforts. [25] Further research is needed to assess the feasibility of task-sharing for cervical cancer screening within the Malawian health system.

## Strengths and limitations

One of the key strengths of this study is the emphasis it has on comprehending women's perspectives and experiences in a rural context. This offers insightful information about the barriers and facilitators of self-sampling for cervical cancer screening. The use of qualitative methods provided a deep and nuanced understanding of the participants' views and contextual factors influencing their acceptance of self-sampling. Furthermore, a rigorous analysis was ensured by the use of the TFA to analyse the data, which improved the understanding and classification of the themes.

However, it is important to note the contextual limitations the findings have as the study was limited to a single district hospital, thus it might not accurately reflect the experiences of women in other regions of Malawi or different cultural contexts. Another limitation of this study is that only female participants were included in the study, and the absence of male viewpoints especially on spousal acceptance would have limited a more comprehensive knowledge of gender dynamics in the adoption of self-sampling. Similarly, the study only included women who had engaged in self-sampling, limiting the ability to explore views from those who declined or had not yet experienced the intervention. Further studies could build on these limitations by using a more varied sample and investigating how male partners influence women's health-seeking behaviours.

## Conclusion

Self-sampling for cervical cancer screening is generally acceptable among women, and it has the potential to provide a favourable alternative to the traditional provider-delivered sampling. Women had positive attitudes towards self-sampling, confidence to conduct the self-sampling procedure, and awareness of the intervention and its benefits. However, its full-scale adoption must be approached with careful consideration. Despite facilitating factors, key barriers such as lack of spousal support, time constraints, and low literacy levels remain. Addressing these challenges is crucial for the successful implementation of self-sampling in cervical cancer screening programs. Cognitive interviews should be conducted to refine IFUs to align with users' literacy levels, while targeted community-based interventions should be designed to enhance male involvement in cervical cancer awareness. Further studies on women's acceptability of self-sampling for cervical cancer screening should consider expanding the scope of this study to include diverse populations and contexts, ensuring broader applicability of self-sampling interventions.

## Supporting information

**S1 Checklist. Responses to COREQ checklist.**
(PDF)

# Acknowledgments

We acknowledge with deep gratitude the late Professor Jon Odland for his invaluable support and vision in strengthening Global Health Research in Africa. His unwavering commitment and leadership were instrumental in facilitating the work on self-sampling for HPV testing for cervical cancer screening in Malawi. This initiative was made possible under his broader efforts to promote equitable and sustainable health research collaborations across the continent. His legacy continues to inspire our work.

# Author contributions

**Conceptualization:** Hussein Hassan Twabi, Vanitha Sivalingam, Dennis Solomon, Chisomo Msefula, David Lissauer, Marriott Nliwasa, Maria Lisa Odland.

**Data curation:** Hussein Hassan Twabi, Wakumanya Sibande, Owen Mhango.

**Formal analysis:** Hussein Hassan Twabi, Wakumanya Sibande, Owen Mhango.

**Funding acquisition:** Hussein Hassan Twabi, Marriott Nliwasa.

**Investigation:** Hussein Hassan Twabi, Owen Mhango, Chikumbutso Chipandwe.

**Methodology:** Hussein Hassan Twabi, Owen Mhango, Takondwa Charles Msosa, Chikumbutso Chipandwe, Madalo Mukoka, Moses Kumwenda, Marriott Nliwasa.

**Project administration:** Hussein Hassan Twabi, Chikumbutso Chipandwe, Marriott Nliwasa.

**Resources:** Hussein Hassan Twabi, Marriott Nliwasa.

**Software:** Hussein Hassan Twabi.

**Supervision:** Chisomo Msefula, David Lissauer, Marriott Nliwasa, Maria Lisa Odland.

**Validation:** Hussein Hassan Twabi.

**Writing – original draft:** Hussein Hassan Twabi, Wakumanya Sibande.

**Writing – review & editing:** Hussein Hassan Twabi, Takondwa Charles Msosa, Madalo Mukoka, Moses Kumwenda, Vanitha Sivalingam, Dennis Solomon, Chisomo Msefula, David Lissauer, Marriott Nliwasa, Maria Lisa Odland.

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
