## [Decision Letter · Decision Letter 0]

17 Jun 2025

PGPH-D-25-01300

Acceptability of Self-sampling for High-risk HPV DNA Testing for Primary Cervical Cancer Screening Among Women in Thyolo, Malawi: A Qualitative Study

Dear Dr. Twabi,

Thank you for submitting your manuscript to PLOS Global Public Health. After careful consideration, we feel that it has merit but does not fully meet PLOS Global Public Health’s publication criteria as it currently stands. Therefore, we invite you to submit a revised version of the manuscript that addresses the points raised during the review process.

We look forward to receiving your revised manuscript.

Kind regards,

Edina Amponsah-Dacosta, Ph.D., MPH

Academic Editor

Journal Requirements:

1. We do not publish any copyright or trademark symbols that usually accompany proprietary names, eg (R), (C), or TM (e.g. next to drug or reagent names). Please remove all instances of trademark/copyright symbols throughout the text, including ® on page 5.

2. In the online submission form, you indicated that [All data are available upon reasonable request to the corresponding author, after publication.].

a. In a public repository,

b. Within the manuscript itself, or

c. Uploaded as supplementary information.

Additional Editor Comments (if provided):

Reviewers' comments:

Reviewer's Responses to Questions

**Comments to the Author**

1. Does this manuscript meet PLOS Global Public Health’s publication criteria ? Is the manuscript technically sound, and do the data support the conclusions? The manuscript must describe methodologically and ethically rigorous research with conclusions that are appropriately drawn based on the data presented.

Reviewer #1: Yes

Reviewer #2: Yes

2. Has the statistical analysis been performed appropriately and rigorously?

Reviewer #1: N/A

Reviewer #2: N/A

3. Have the authors made all data underlying the findings in their manuscript fully available (please refer to the Data Availability Statement at the start of the manuscript PDF file)?

Reviewer #1: Yes

Reviewer #2: Yes

4. Is the manuscript presented in an intelligible fashion and written in standard English?

Reviewer #1: Yes

Reviewer #2: Yes

5. Review Comments to the Author

Reviewer #1: The authors present qualitative work describing Malawian women’s experiences with self-sampling in preparation for the development and testing of a self-sampling device and point-of-care HPV test.

Lines 73-75: The authors provide the global number of incident cervical cancer cases and deaths, not the numbers limited to Sub-Saharan Africa.

Lines 96-98: The authors reference emerging POC HPV tests, referencing an article from 2015. There is still no currently approved POC HPV test, and many countries have already moved toward recommendations of HPV testing in their national strategy. The authors may be referring to lower cost testing that requires less complex laboratory infrastructure or trained personnel.

Lines 111-118: It’s not clear how self-sampling, which would ostensibly increase the number of HPV tests done, would alleviate the burden on an overextended diagnostic system or address the follow-up challenges still inherent for results provision and appropriate management.

It’s not clear whether the participants in the study had seen or used the novel device for screening as part of/in addition to the self-sampling for Xpert HPV. Given the complexity of the process (requiring a self-lavage), it would be valuable to understand how women experienced self-collection with that device, in comparison to using a flocked swab. Is that what is meant by “utilized the intervention” which is mentioned several times in the paper?

What are the screening guidelines for Malawi? In most settings, a 19 yr old, HIV uninfected, would not be offered screening.

Table 2: Please adjust the formatting so that there are no “hanging” or wraparound words.

Lines 337-343: What process of integrated self-sampling and self-screening were the IDIs probing/the women describing here?

There are a few limitations that bear mentioning in the limitations section; the authors only spoke with women who had already engaged in self-sampling. This limited their ability to understand barriers to acceptability. In addition, while many studies have shown widespread acceptability of self-screening, the need to ensure appropriate follow-up is crucial, especially in contexts where women have screened outside a health care system or without a provider’s counseling. Did the researchers consider asking about how women would respond to or prefer to receive results or navigate follow-up care?

Reviewer #2: I commend the authors for their efforts in this manuscript. The manuscript covers the acceptability of self-sampling for high-risk HPV DNA testing for primary cervical cancer screening among women in Malawi. The title is accurate, the abstract is well-written, and the manuscript is well-structured overall. However, there are a few observations I will like the authors to address.

Background: Additional and recent references on the prevalence and incidence of cervical cancer worldwide and in Malawi should be included. The focus should be on the general population, since the study was conducted among the general population of women and not women living with HIV (Line 81).

Methods: The methods section should be more structured to enhance readability; the study design, sampling technique, sample size, etc should come first before the intervention. The data management and analysis plan, and ethical considerations should end the section. Provide a justification for the sample size e,g. was saturation attained at the 10th participant? In addition, was a sampling matrix prepared as an inclusion criteria before recruiting study participants? Also, include information about incentive if any given to the participants. What other permissions and approvals were obtained? The abbreviation “TFA” should be defined at first mention (Line 159). Why was ethicality and opportunity cost excluded from the analysis? (Line 225). Supporting documents should include responses from the COREQ checklist (Line 234).

Result: The ages of the participants looks like they were randomly selected and not purposive. What was the criteria for the purposive selection of the study participants (Table 2)? Please note that manuscript writing is not a dissertation. Hence, the characteristics of the study participants can be shortened and referred to table 2. Avoid repeating points. Privacy has been explained in the earlier paragraph. If there is no direct quote for comfort, consider merging privacy and comfort as one (Line 287-294).

You can focus on the quote directly by excluding “in addition” (Line 303). I suggest that a conceptual framework be drawn to link all the factors in the TFA construct identified in the study. Line 310-311 has already highlighted time as a burden. It is again raised in the next paragraph, Line 320-323. Use the past tense of perceive (Line 410).

Discussion: For a more robust discussion, consider including the following references:

1. Bakiewicz A, Rasch V, Mwaiselage J, Linde DS. “the best thing is that you are doing it for yourself” - Perspectives on acceptability and feasibility of HPV self-sampling among cervical cancer screening clients in Tanzania: A qualitative pilot study. BMC Womens Health. 2020;20(1):1–9.

2. Sormani J, Kenfack B, Wisniak A, Datchoua AM, Makajio SL, Schmidt NC, et al. Exploring factors associated with patients who prefer clinician-sampling to HPV self-sampling: A study conducted in a low-resource setting. Int J Environ Res Public Health. 2022;19(1):1–11.

3. Saidu R, Moodley J, Tergas A, Momberg M, Boa R, Wright T, et al. South african women’s perspectives on self-sampling for cervical cancer screening: A mixed-methods study. South African Med J. 2019;109(1):47–52.

4. Pierz AJ, Ajeh R, Fuhngwa N, Nasah J, Dzudie A, Nkeng R, et al. Acceptability of Self-Sampling for Cervical Cancer Screening Among Women Living With HIV and HIV-Negative Women in Limbé, Cameroon. Front Reprod Heal. 2021;2(January):1–8.

6. PLOS authors have the option to publish the peer review history of their article (what does this mean? ). If published, this will include your full peer review and any attached files.

**Do you want your identity to be public for this peer review?** For information about this choice, including consent withdrawal, please see our Privacy Policy .

Reviewer #1: No

Reviewer #2: **Yes: ** Andrew-Bassey, Uduak Ima

---

## [Decision Letter · Decision Letter 1]

15 Aug 2025

PGPH-D-25-01300R1

Acceptability of Self-sampling for High-risk HPV DNA Testing for Primary Cervical Cancer Screening Among Women in Thyolo, Malawi: A Qualitative Study

Dear Dr. Twabi,

Thank you for submitting your manuscript to PLOS Global Public Health. After careful consideration, we feel that it has merit but does not fully meet PLOS Global Public Health’s publication criteria as it currently stands. The reviewers have raised further points for your attention. Therefore, we invite you to submit a revised version of the manuscript that addresses the points raised during the review process.

We look forward to receiving your revised manuscript.

Kind regards,

Edina Amponsah-Dacosta, Ph.D., MPH

Academic Editor

Journal Requirements:

Additional Editor Comments (if provided):

Reviewers' comments:

Reviewer's Responses to Questions

**Comments to the Author**

1. If the authors have adequately addressed your comments raised in a previous round of review and you feel that this manuscript is now acceptable for publication, you may indicate that here to bypass the “Comments to the Author” section, enter your conflict of interest statement in the “Confidential to Editor” section, and submit your "Accept" recommendation.

Reviewer #1: (No Response)

Reviewer #2: All comments have been addressed

2. Does this manuscript meet PLOS Global Public Health’s publication criteria ? Is the manuscript technically sound, and do the data support the conclusions? The manuscript must describe methodologically and ethically rigorous research with conclusions that are appropriately drawn based on the data presented.

Reviewer #1: Yes

Reviewer #2: Yes

3. Has the statistical analysis been performed appropriately and rigorously?

Reviewer #1: N/A

Reviewer #2: N/A

4. Have the authors made all data underlying the findings in their manuscript fully available (please refer to the Data Availability Statement at the start of the manuscript PDF file)?

Reviewer #1: Yes

Reviewer #2: Yes

5. Is the manuscript presented in an intelligible fashion and written in standard English?

Reviewer #1: Yes

Reviewer #2: Yes

6. Review Comments to the Author

Reviewer #1: Thank you for the opportunity to review this revised manuscript. The authors have made changes that result in an overall improvement in the manuscript, however there are some issues raised by both reviewers that were not adequately addressed in the manuscript (just in the responses).

In the methods section, the authors need to clearly state that they recruited women who had undergone cervical cancer screening using Xpert HPV with the Delphi Vaginal self-sampler. Both reviewers requested clarification on that, and the revisions to describe purposive sampling and inclusion are not adequate. In addition, were in-depth interview participants already participants in a larger implementation trial, or had the Delphi Vaginal self-sampler been introduced as standard for self-collection?

Overall, the paper lacks a discussion of how women would access follow-up or treatment after self-sampling, as screening alone has no public health benefit without appropriate action for women who screen positive. The expanded discussion paragraph starting on line 524 may be an appropriate place to add this as the authors may be alluding to this, and/or in the limitations or conclusion paragraph.

Reviewer #2: Just a slight correction for introductory part of the manuscript; Line 70-84: Use the funnel approach; worldwide, sub-Saharan Africa, and then Malawi.

Also, incentives given to participants and permissions received before conducting the study should be included in the ethical consideration section of the manuscript.

7. PLOS authors have the option to publish the peer review history of their article (what does this mean? ). If published, this will include your full peer review and any attached files.

**Do you want your identity to be public for this peer review?** For information about this choice, including consent withdrawal, please see our Privacy Policy .

Reviewer #1: No

Reviewer #2: **Yes: ** Andrew-Bassey, Uduak Ima

---

## [Decision Letter · Decision Letter 2]

9 Sep 2025

Acceptability of Self-sampling for High-risk HPV DNA Testing for Primary Cervical Cancer Screening Among Women in Thyolo, Malawi: A Qualitative Study

PGPH-D-25-01300R2

Dear Dr. Twabi,

We are pleased to inform you that your manuscript 'Acceptability of Self-sampling for High-risk HPV DNA Testing for Primary Cervical Cancer Screening Among Women in Thyolo, Malawi: A Qualitative Study' has been provisionally accepted for publication in PLOS Global Public Health.

Best regards,

Edina Amponsah-Dacosta, Ph.D., MPH

Academic Editor

Reviewer #2:

Reviewer Comments (if any, and for reference):

Reviewer's Responses to Questions

**Comments to the Author**

1. If the authors have adequately addressed your comments raised in a previous round of review and you feel that this manuscript is now acceptable for publication, you may indicate that here to bypass the “Comments to the Author” section, enter your conflict of interest statement in the “Confidential to Editor” section, and submit your "Accept" recommendation.

Reviewer #2: All comments have been addressed

2. Does this manuscript meet PLOS Global Public Health’s publication criteria ? Is the manuscript technically sound, and do the data support the conclusions? The manuscript must describe methodologically and ethically rigorous research with conclusions that are appropriately drawn based on the data presented.

Reviewer #2: Yes

3. Has the statistical analysis been performed appropriately and rigorously?

Reviewer #2: N/A

4. Have the authors made all data underlying the findings in their manuscript fully available (please refer to the Data Availability Statement at the start of the manuscript PDF file)?

Reviewer #2: Yes

5. Is the manuscript presented in an intelligible fashion and written in standard English?

Reviewer #2: Yes

6. Review Comments to the Author

Reviewer #2: The authors have extensively worked on the paper, and requires no further review.

7. PLOS authors have the option to publish the peer review history of their article (what does this mean? ). If published, this will include your full peer review and any attached files.

**Do you want your identity to be public for this peer review?** For information about this choice, including consent withdrawal, please see our Privacy Policy .

Reviewer #2: No
